# Trends in hydroclimate extremes: How changes in winter affect water storage and baseflow

Tejshree Tiwari[1], Hjalmar Laudon[1]

[1]Department of Forest Ecology and Management, Swedish University of Agricultural Sciences, SE-901 83 Umea, Sweden

*Correspondence to*: Tejshree Tiwari (Tejshree.Tiwari@slu.se)

**Abstract.** Northern ecosystems are undergoing accelerated climate warming, with average temperature increases exceeding the global mean. In snow-dominated catchments, where cold-season conditions are essential for sustaining streamflow across subsequent seasons, substantial uncertainty persists regarding the impacts of future warming on catchment water storage and runoff dynamics. Here, we utilized 40 years of hydrological data from the boreal Krycklan catchment, set within a 130-year

climate record from a nearby station, to evaluate how 27 extreme climate indices can capture changes and trends in water storage and stream low flow during winter and summer. Our results show that annual temperatures have risen by 2.2 °C over the past four decades, with even more pronounced seasonal impacts. Notably, six winter extreme indices and two summer indices revealed distinct trends. We found that warm winters have led to increased winter stream runoff but reduced summer runoff. Predictive modelling indicated that the accumulated freezing degree days (AFDD<0) were the strongest predictor of

minimum winter flow, while a combination of AFDD<0 and maximum summer temperature (MaxTmax) best explained variations in minimum summer flow. Furthermore, analysis of streamflow partitioning using water isotopes and the seasonal origin index (SOI) over the past 22 years revealed an increasing winter precipitation signal in winter runoff, accompanied by a declining contribution to summer streamflow. Together, these findings demonstrate that warm winters are fundamentally altering catchment-scale water storage and flow partitioning, with important implications for water availability and

ecosystem functioning during the growing season in boreal landscapes.

## 1 Introduction

Over the past decades, high-latitude areas have undergone a warming trend exceeding the global average, with the region continuing to warm at a rate more than twice as fast as the rest of the world (Druckenmiller et al. 2021, Rantanen et al. 2022). The boreal zone is particularly sensitive to changes in climate (Seidl et al. 2020, Fu et al. 2023, Ali et al. 2024) as its thermal

regime is strongly affected by the snow cover. Hence, temperature changes are likely to affect snow accumulation and the timing of melt (Kim et al. 2012, Peng et al. 2013, Friesen et al. 2021, Bouchard et al. 2024), which are mechanisms important for regulating the length of the growing season and plant phenology (Easterling 2002, Cleland et al. 2007, Way 2011). Consequently, shifts in the timing of the onset and end of winter are among the most fundamental anticipated effects of climate change in northern latitudes, which can increase the risk of severe, and in some cases irreversible ecological impacts. Yet, how

changes in seasonal cycles, temperature and moisture regimes feedback on terrestrial ecosystems, biogeochemical cycles and hydrological balance remain largely unknown.

Winter conditions define the timing and magnitude of hydrological processes within northern catchments particularly as seen in river runoff (Barnett et al. 2005, Blöschl et al. 2017, Murray et al. 2023, Hrycik et al. 2024). For instance, temperatures below freezing are conducive for precipitation to accumulate as snow and ice, which is made available for groundwater recharge during melting. The freezing period is therefore a key determinant of the amount of water stored in catchments based on temperature departure below zero, but also on the duration of the below-zero period and the subsequent melting rate (Simons 1967, Stieglitz et al. 2001, Nygren et al. 2020). Any changes to the period of below freezing will ultimately affect the duration of the cold period with consequences for the water cycle leading to alterations in the flow regime (Blöschl et al. 2017). Already, observed changes in the winter period have led to significant trends in ice duration, freezing period and break-up dates based on long-term data from Sweden and Finland (Arheimer and Lindström 2015, Hallerbäck et al. 2022). Such results have consequential effects on soil freeze/thaw cycles, and lake and river ice dynamics (Kim et al. 2012, Peng et al. 2013) across the northern hemisphere. However, despite this growing body of evidence about warm winters, the implications on water storage and seasonal runoff dynamics are still elusive.

Changes in river runoff are essential factors for assessing the impact of climate change on the hydrological cycle as the discharge is highly dependent on precipitation (P), evapotranspiration (ET), and changes in water storage (White et al. 2007). One of the most prominent effects of changes in water flux can be seen during the low flow periods, which for a given catchment largely is regulated by the amount of water stored in the catchments. In high latitude and altitude regions with long winters, mid-winter baseflow quantities are primarily reliant on the factors conducive to winter snowmelt or direct contributions from rain on snow. In contrast, summer low flows are often more dependent on water storage in soil and groundwater, often dominated by recharge occurring during snowmelt. Despite the host of studies that have demonstrated that snow melts earlier in years with less winter snow accumulation (Venäläinen et al. 2020, Irannezhad et al. 2022, Hrycik et al. 2024), resulting in decreasing summer baseflow trends (Murray et al. 2023), clear evidence on how winter climate affects runoff, and consequently catchment recharge is still lacking (Tiwari et al. 2018, 2019). Understanding how such changes in winter conditions will affect subsequent seasonal runoff patterns is largely reliant on identifying techniques that can detect changes in the duration, magnitude and intensity of the freezing period, and their application towards understanding how water recharge and storage is affected (Dierauer et al. 2018, Blahušiaková et al. 2020).

A powerful technique for assessing changes in stream flow involves using the differences in the water isotopic signatures of $\delta^{18}O$ in precipitation across seasons (Allen et al. 2019). The seasonality of isotopic signals in precipitation presents the possibility of tracing the fraction of water arriving in winter that can either be stored in the catchment or become streamflow directly during the current or during succeeding seasons. Similarly, the SOI can be used to infer the fraction of winter

precipitation that becomes stream runoff during the summer. Using isotopic signals to understand the water partitioning in catchments offers a possibility to trace the role of winter precipitation contributions to stream runoff that is not possible by using hydrometric measurements only. This can be done by adapting the seasonal origin index (SOI) to implicitly test the proportion of winter precipitation versus summer precipitation in stream water with $\delta^{18}O$ isotopes using the methods outlined in Allen et al. (2019). The results from the SOI analysis can then be used to test whether increases in winter runoff and decreases in summer baseflow are the results of changes in winter precipitation.

The techniques of identifying when changes in climate time series occur usually involve using models to detect the statistical departures from historical baselines (Alexander et al. 2006, Reeves et al. 2007, Wilmking et al. 2020). However, monitoring trends in seasonal variables has become increasingly challenging, as traditional definitions of seasonality are insufficient to reflect changes associated with fluctuating weather conditions. For instance, the most commonly used definitions (e.g. astronomical and meteorological) are static in both time and space (Trenberth 1983, Allen. and Sheridan 2015). Considerable changes to transition periods (spring and fall) between the warmest seasons in the summer to the coldest season in the winter are often not adequately defined based on their spatiotemporal variability (Huschke 1959). As such, static definitions of seasons are insufficient to properly characterize seasonal timing and length, which are also likely to change in the future. Currently, many indicators of seasonality changes are based on temperature-related impacts on ecosystems, and hence they provide estimates of changes in the seasonal timing of the growing season. Minimum, mean, and maximum daily temperature have been taken as the most important temperature characteristics in the analysis of climate change (Moberg et al. 2006, Cohen et al. 2014, Cassou and Cattiaux 2016). Various indices have been used in the literature to divide seasons, such as those based on temperature (Alexander et al. 2006, Hekmatzadeh et al. 2020) and phenology (Schwartz and Crawford 2001, Cleland et al. 2007, Peng et al. 2013), moving average smoothing techniques to identify the time that temperature rises above or fall below long-term mean (Blöschl et al. 2017, Park et al. 2021), or defined thresholds such as the 75th and 25th percentile to identify the coldest and warmest periods (Zschenderlein et al. 2019). These studies suggest that the onset and offset of seasons depend on the geographical location of the study region and its specific purpose where a variable threshold should be used for determining the start and end dates.

This study integrates long-term hydro-climatic trend analysis, seasonal extreme indices, and water isotope-based seasonal origin tracing to assess how changes in winter climate influence inferred water storage and seasonal runoff dynamics in a well-studied boreal catchment. Although the effects of snowmelt and warming on streamflow have been widely investigated, the direct role of winter climate in controlling water storage and baseflow remains poorly quantified. To address this gap, a combined approach was applied using historical trend analysis, indices of climate extremes, and stable isotope data to evaluate the effects of winter climate on seasonal streamflow. The first objective was to identify possible significant changes in temperature trends using a 130-year historical dataset and assess their relationship to more recent changes captured in a 40-year on-site time series. Seasonal climate extremes were then derived from the last 30 years where we have high-

resolution data to identify extremes across the seasons. The second objective focused on quantifying the influence of these climate extremes on key hydrological processes during winter and summer, using regression analysis to identify dominant drivers of seasonal runoff. The final objective involved the application of the seasonal origin index (SOI), based on δ¹⁸O

100   isotope signatures, to quantify the relative contribution of winter precipitation to streamflow during winter and summer and to verify consistency with observed hydroclimatic trends.

2 Methodology

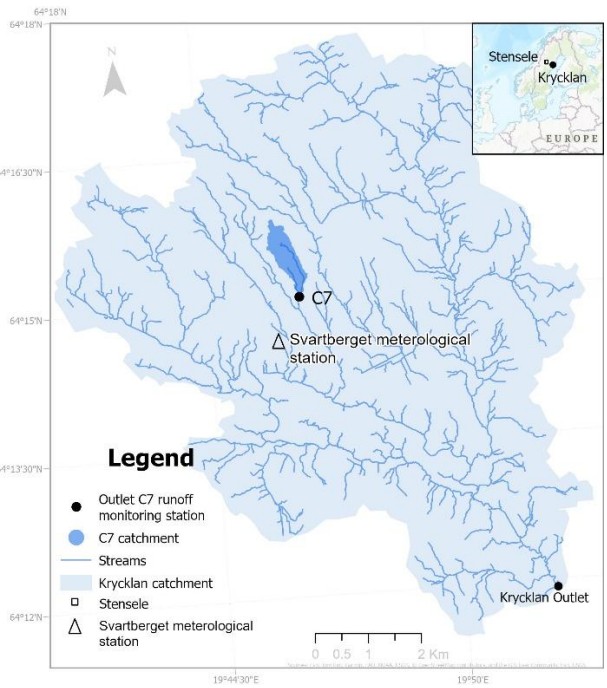

**Figure 1 Map showing the Krycklan catchment, the location of the Svartberget meteorological tower in relation to the C7 catchment, the runoff monitoring station and the nearby Stensele meteorological station.**

**The Krycklan Catchment and Svartberget**

This study focuses on the Krycklan catchment, which is a research infrastructure that has been monitored since the 1980s

(Laudon and Sponseller 2018). It is located in the boreal zone of northern Sweden, approximately one hour from the city of Umeå and the Baltic Sea. Nested within the Krycklan catchment is the Svartberget catchment (C7), which has the longest high-quality monitoring record of runoff in the region (Fig. 1) (Laudon et al. 2021). The 47 ha of C7 is dominated by forest (81%), primarily Norway spruce (*Picea abies*) and Scots pine (*Pinus sylvestris*) on till soils (81%) and peatland (19%). The location of Svartberget represents the conditions in Scandinavian boreal forests and is part of a network of stations within the Integrated

Carbon Observation System (ICOS) - a European research infrastructure established to quantify and understand the greenhouse gas balance of the European continent and adjacent regions (Chi et al. 2020).

**Meteorological measurements and data**

Precipitation and temperature data from the Svartberget field station (Fig. 1) was obtained from the SITES data portal (https://data.fieldsites.se/portal/). Precipitation (mm) was determined using daily (00-24) accumulated rain and snowfall

amounts from manual measurement using a standard Swedish Meteorological and Hydrological Institute (SMHI) gauge with a windshield located 1.5 m above the ground (Climate monitoring program at SLU experimental forests and SITES Svartberget) (Laudon et al. 2013). The data available for Svartberget included two time periods (1982-2022) which consisted of daily mean precipitation and temperature used to detect long-term trends and (1992-2022) with daily temperature, average daily minimum (min.) and average daily maximum (max.) temperature needed for the extreme climate change indices

detection. Two data sets were used in this study because the earlier dataset 1982-1992 only provided one value for daily mean while 1992-2022 provided 10-minute recordings of temperatures from which daily minimum and maximums could be determined.  Average daily minimum (Tmin) and maximums (Tmax) represent the coldest and warmest temperatures recorded in a day (Fig. S1). Both Tmin and Tmax were used in the extreme indices to identify the coldest (minTmin) and warmest (maxTmax) in the season.


In addition to the Krycklan data, we included a longer-term dataset (1891–2004) from the SMHI meteorological station in Stensele, located approximately 150 km west of the Krycklan catchment (Fig. 1). The Stensele dataset was used to place observed trends from the past 40 years in Krycklan into a longer-term climatic context. Temperature records from Stensele and Svartberget show strong agreement during their overlapping period (1982–2004), with an $r^2$ of 0.92 and a root mean square

error (RMSE) of 0.05 based on daily average temperatures (Fig. S2). Minor systematic biases were identified: maximum temperatures were slightly higher in Svartberget, whereas minimum temperatures were slightly lower in Stensele. However, these differences were small (<3%) and did not affect the overall trends in the time series, which were the primary focus of this study. Therefore, the Stensele dataset was considered a reliable proxy for assessing historical climatic trends in the region.

**Runoff measurements and data**

Runoff measurements from the Svartberget catchment (C7) were done using a field-based recording of hourly stage height and established rating curves to calculate the daily discharge from a weir located in a heated hut. The rating curve was established using the salt dilution technique and bucket-method measurements (Laudon et al. 2004). Occasional missing data (4%) were gap-filled using the HBV model (Karlsen et al. 2016, Karimi et al. 2022). The data available extended across 40 years from 1982-2022. The entire time series was used to show the long-term trends in minimum runoff (Qmin); however, the modelling

was done using a subset of the data set (1992-2022) for the seasonal analysis to synchronise with the extreme indices.

**Seasonal definitions**

Seasons were separated according to the thermal threshold definition (Contosta et al. 2020), where winter is a consistent frozen period when the air temperature is below 0°C for more than seven consecutive days (Fig. S3). We used the SMHI 10°C threshold for the definition of summer, which is the period when the daily mean air temperature was above 10°C for more than
five consecutive days. Summer ended when the mean air temperature fell below 10°C for more than five consecutive days. The spring period between winter and summer was classified as the period when air temperature was above 0°C for more than seven consecutive days but less than 10°C for more than seven consecutive days. The autumn season was not used in this study; however, we checked the effects of autumn precipitation and runoff on mid-winter runoff and winter Qmin which did not show any significant effects (Fig. S4).

**Extreme climate indices**

In total 27 climate change indices were used in this study, which were developed by the World Meteorological Organization (WMO) Expert Team of Climate Change Detection and Indices (Donat et al. 2020). These indices assess various aspects of temperature and precipitation variability including i) intensity, ii) duration, and iii) frequency of events. The indices identified were then determined for each seasonal block (winter, spring and summer (Table 1)) and used in the regression analysis to
understand the relation to runoff. To address inhomogeneity in the dataset, we used the bootstrap technique to test if the trends in climate indices would vary depending on the window use i.e. 3 days, 5 days, 7 days) based on the description of the climate index (https://etccdi.pacificclimate.org/list_27_indices.shtml).

**Temperature and precipitation extreme climate indices**

Temperature intensity was assessed using seven variables from the daily average, minimum and maximum as follows: (i) the
coldest daily maximum temperature (Min Tmax), (ii) the coldest daily minimum temperature (Min Tmin), (iii) the warmest daily maximum temperature (Max Tmax), (iv) the warmest daily minimum temperature (Max Tmin), (v) the mean difference between daily maximum and daily minimum temperature (diurnal temperature range), (vi) the number of days when Tmax < 0°C (icing days), and (vii) the accumulated degree days below 0°C (AFDD<0). The duration of extreme periods of the seasons was categorised as (i) the number of days with at least six consecutive days when Tmin < 10th percentile (cold spell), and (ii)
the number of episodes with at least six consecutive days with Tmax > 90th percentile (warm spell). The frequency of events within each season was identified as; (i) the percentage of days when Tmax < 10th percentile (cool days), (ii) the percentage of days when Tmin < 10th percentile (cool nights), (iii) the percentage of days when Tmax > 90th percentile (warm days), (iv) the percentage of days when Tmin > 90th percentile (warm nights), and (v) the number of days when Tmin < 0°C (frost days; Table 1).

The intensity, duration, and frequency of seasonal precipitation patterns were determined to also highlight extremes across the years. Precipitation intensity was measured using; (i) maximum 1-day precipitation total, (ii) maximum 5-day precipitation

total, (iii) sum of daily precipitation > 95th percentile (wet days), and (iv) sum of daily precipitation > 99th percentile (very wet days) while duration of precipitation events were done using (i) the maximum number of consecutive wet days (precipitation >1 mm) and (ii) the maximum number of consecutive dry days (precipitation). It should be noted that W and S are used in the regression models to distinguish between winter and summer variables.

**Analysis: Trend detection and regression analysis**

Trends in extreme climate indices were assessed using the trend detection package in R (R Development Core Team, 2021), applied to long-term (1982–2022) datasets of average daily temperature, precipitation, and runoff. Analyses were conducted for both annual and seasonal datasets (winter, spring, and summer) across 27 indices, examining trends in mean values and variability (standard deviation). The Mann-Kendall test was used to determine trend direction and p-values for significance. To identify which significant climate indices best explained minimum seasonal runoff (Qmin), stepwise linear regression was performed using Minitab® Statistical Software (2021). For winter Qmin, significant winter temperature and precipitation indices were used, while summer Qmin was predicted based on significant indices from winter, spring, and summer.

**$\delta^{18}O$ and Seasonal Origin index**

As a verification test for understanding the contributions of seasonal precipitation to annual runoff, the analysis of $\delta^{18}O$ isotopes was carried out. Samples were collected at regular intervals from 2002 to 2022 in precipitation (n=1930) and stream water (n=821). Precipitation sampling was done manually using national standard precipitation gauges with a windshield where samples were collected in dark glass bottles with a hermetic lid to minimize evaporation during storage. Stream water samples were collected weekly with more frequent sampling during the snowmelt season in similar bottles as precipitation samples. During the winter, a heated weir house enables sampling throughout the frozen season. Both precipitation and stream samples were analysed using a Picarro cavity ringdown laser spectrometer (L1102-i and L2130-i after September 2013) and the vaporizer module (V1102-i and later the A0211) (See Peralta-Tapia et al. (2016) for more details). Calibration of the isotopic signatures of water was done using internal laboratory standards calibrated against three International Atomic Energy Agency (IAEA) official standards, the Vienna Standard Mean Ocean Water (VSMOW), the Greenland Ice Sheet Precipitation (GISP) and the Standard Light Antarctic Precipitation (SLAP) (Coplen 1995).

The $\delta^{18}O$ data was then used in the seasonal origin index (SOI) analysis to test whether winter or summer precipitation is overrepresented in seasonal streamflow (Allen et al. 2019). The SOI is a technique that can be used to identify the prevalence of winter versus summer precipitation sources in stream water due to the inherent differences in the winter and summer precipitation signals. This can be done using the deviation of annual average discharge from annual precipitation and scales the deviation by the strength of the seasonal signals using eq.1

$$SOI_{\overline{Q}} = \frac{\delta_{\overline{Q}} - \delta_{\overline{P}}}{\delta_{P_S} - \delta_{\overline{P}}} \quad \text{if } \delta_{\overline{Q}} > \delta_{\overline{P}} \qquad \text{eq.1}$$

$$\frac{\delta_{\overline{Q}} - \delta_{\overline{P}}}{\delta_{\overline{P}} - \delta_{P_W}} \;\; \text{if } \delta_{\overline{Q}} < \delta_{\overline{P}}$$

Where $\delta_{P_S}$, $\delta_{P_W}$, $\delta_{\overline{P}}$ and $\delta_{\overline{Q}}$ represents the $\delta^{18}O$ values of summer, winter, and volume-weighted annual precipitation in the C7 catchment and annual volume-weighted mean streamflow $\delta^{18}O$ respectively. To determine the SOI of winter and summer, we calculate the SOI of individual streamflow samples $\delta_{\overline{Q}}$ using their individual isotope ratios. The January samples were used to represent the winter, and the July samples were used to represent the summer isotopic signals. Linear regression was then used to show the proportion of winter precipitation in winter and summer baseflow where positive $SOI_{\overline{Q}}$ (closer to 1) in the summer

indicates a larger fraction of summer precipitation in streamflow. Similarly, increasing negative $SOI_{\overline{Q}}$ (closer to -1) during the winter means greater contributions from winter precipitation to stream flow.

**Results**

**Trends in temperature, precipitation and runoff in Svartberget**

Over the last four decades, the increase in temperature has accelerated by 2.2 °C where the long-term average daily temperature

trend changed from 1°C in 1980 to 3.2 °C in 2022 (Fig. 2A). Placing these trends into a much longer perspective by using time series that extended to 1892 from the nearby SMHI-Stensele meteorological station, we note that the increasing trends observed in the Svartberget are part of the much longer-term temperature trend that extends over 100 years. Looking back at the 30-year normal period trends over this period, we can observe that at the beginning of the century, 1892-1922 annual average air temperature was much colder (-0.29 °C) than the most recent 30-year normal period (1993-2022) which was warmer (2.5 °C)

(Fig. 2A).

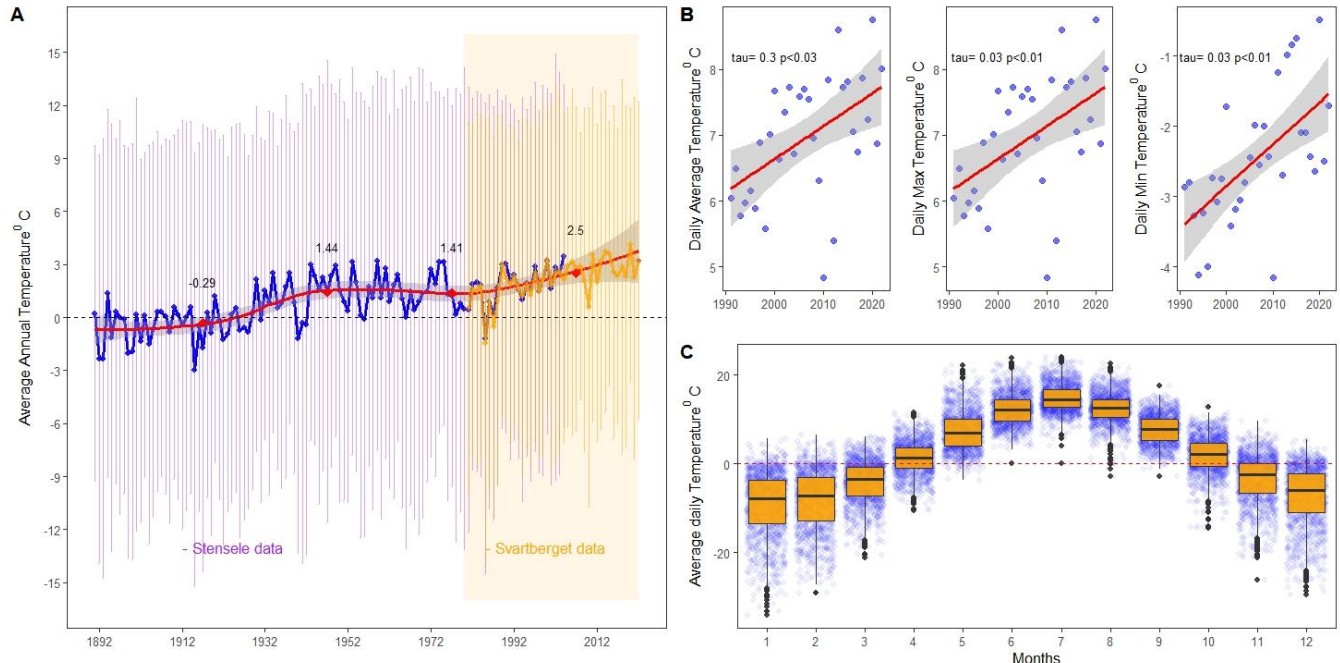

**Figure 2 Trend in long-term temperature in the Krycklan catchment showing; (A) the relation to the much longer-term time series from the SMHI- Stensele data (1891-2004), (B) trends in average daily air temperature (Avg.), maximum (Max.) daily air temperature and (C) minimum (Min.) daily temperatures, and variability in daily temperatures in the months, across 30 years from 1992-2022. In panel A, the red symbols indicate the average within a 30-year period from 2022 backwards while purple and orange error bars represent the standard deviation in the Stensele and Svartberget datasets, respectively. The grey shaded area in panel B**
**represents the standard error in the datasets and the blue jitter dots in panel C represent daily average temperatures in Svartberget.**

A closer look at the last 30 years of variability in temperatures in the Krycklan catchment (1992-2022) showed annual average temperatures ranging from 0.5 to 5 °C across the years increasing from 1.5 °C in 1992 to 3.1 °C by 2022 (Fig. 2B) when observing the long-term trend line. Trends in average daily temperatures showed significant increases in observed annual averages (0.3, p<0.03), average daily maximums (0.3, p<0.01) and average daily minimums (0.3, p<0.01) during the period

1992-2022 (Fig. 2B, Table S1). Annual average maximum temperatures ranged from 4.8 to 8.7 °C with an increasing trend from 6.4 °C in 1991 to 7.8 °C in 2022 also indicating an increase in maximum temperature of 1.4 °C across the 30 years. Warmest periods occur during month 7 (July) when the temperature ranges from 17.4° to 25.2 °C (Fig. 2C). The annual average minimum temperature ranges from -4.1 to -0.4 °C with the coldest average monthly temperatures occurring in either January or February, ranging from -20.6 to -7.6 °C (Fig. 2C). There is a significant increase in minimum temperatures from 1992 to

2022 (-3.4 °C to -1.6 °C) indicating an increase of 1.8 °C during the coldest months (Fig. 2B). These changes in minimum temperatures suggest that the winter seasons on average are becoming warmer.

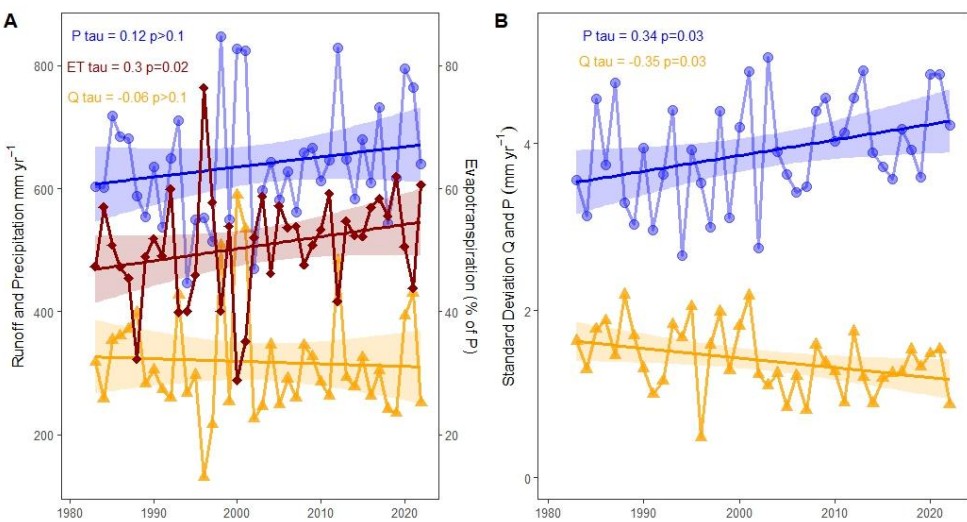

**Figure 3 Long-term trends in total annual Precipitation (P) and total annual runoff (Q) from the Krycklan catchment across 40 years showing the differences in the percentage of precipitation (P) from runoff (Q) that represents evapotranspiration (ET) (panel A). The variability as measured by the standard deviation in runoff (Q) and precipitation (P) during the period is shown in panel B. The shaded areas represent the standard error in the datasets.**

The long-term trends 1982-2022 showed an increase in total annual precipitation while total annual runoff showed a decrease when fitted with a linear regression (Fig. 3A). A closer look at the variability in both the total annual precipitation and runoff across the time series indicated that trends were variable across the years decreasing in the period 1982-1997 followed by an increase towards 2022 as shown with the loess regression (Fig. S5, Table S1). It should be noted that this variability as measured by the standard deviation in runoff has decreased by 0.5 mm day$^{-1}$ from 1.6 mm day$^{-1}$ to 1.1 mm day$^{-1}$ while the variability in precipitation increased significantly from 3.5 to 4.2 mm day$^{-1}$ during the same period (Fig. 3B). Annually, on average the highest total monthly precipitation occurred in July (86 mm on average across the years) while the lowest occurred in April (28 mm). The annual average precipitation was 650 mm per year with the lowest records occurring in 1994 and 2002 (446 and 470 mm respectively) (Fig. 3A) when July/August precipitation was less than 30% of the long-term July/Aug values. The average total annual runoff was 318 mm year$^{-1}$ with the lowest runoff occurring in February (5.8 mm month$^{-1}$) and the highest in May (spring flood) with 96 mm month$^{-1}$ accounting for 30% of the total annual runoff. While no significant trends in total annual precipitation and runoff could be detected (tau= 0.12 p>0.1, tau=-0.06, p>0.1, for P and Q respectively), ET during the same period calculated from the differences between precipitation and runoff showed a generally increasing annual trend (0.3, p=0.02) (Fig. 3A).

**Variability in winter and summer climate variables**

The significant changes in minimum and maximum annual temperatures prompted further investigations into the variability of temperature and precipitation during the coldest (winter) and warmest (summer) seasons. The temperature trends in Svartberget during the winter and summer are consistent with the longer-term time series from the SMHI-Stensele dataset

(Fig. 4A and B). The analysis showed significant increases in average winter temperatures (tau = 0.12, p = 0.14) where daily averages ranged from -30.7 °C to 6.1 °C with the coldest winters occurring in 1986 and 2010 (with average winter temperatures of -11 and - 9.5 °C) while the warmest winters in 2014 and 1992 (average winter temperatures -3.7 °C and 3.2 °C) (Fig. 4C). The duration of the winter period changed from 171 days to 140 days over the 30 years showing a loss of 31 days with the start of winter shifting 16 days later and ending 15 days earlier (Fig. 4D and E). Summers on the other hand, while showed

warmer average temperature trends (0.12, p=0.14), the analysis indicated that the season was getting longer (22 days) moving from 80 days in early 1982 to 102 days in 2022 (Fig. 4D and F). The start and end of summers were shifting on average eleven days earlier and later respectively (Fig. 4F). Total precipitation during winter has decreased but increased during the summer, however, neither of these trends was significant (Table. 1).

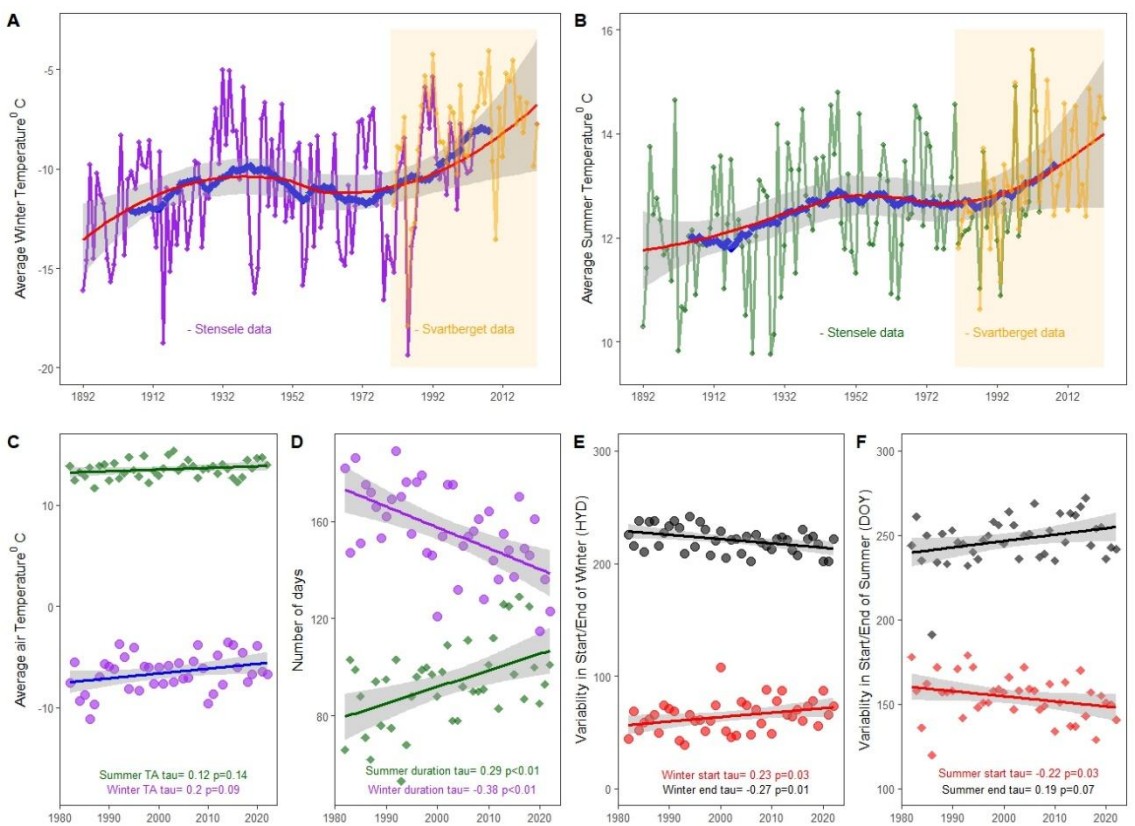

**Figure 4 Variability in seasonal temperature across 40 years in the Krycklan catchment (orange box) showing changes in temperature during the winter (A) and Summer (B) in relation to the SMHI Stensele longer-term dataset (1891-2004). The variability in annual air temperature (C) and duration (D) is represented by purple symbols for the winter period and green symbols for the summer period. Changes in the start (red) and end (black) of the winter (circle) and summer (diamonds) are depicted in panels E and F, respectively. Standard errors in the datasets (A, B, D, E, F) are shown as grey shading.**


Table 1 Extreme climate indices used in the analysis of 30 years of climate data from 1992-2022 showing the description of the indices and the results of the Mann Kendall trend test (tau) and significant (p values). A detailed description of each index can be found at the World Meteorological Organization Expert Team of Climate Change Detection and Indices (http://etccdi.pacificclimate.org/list_27_indices.shtml; commissioned by the World Meteorological Organization Commission for Climatology ((CCl)/CLIVAR/ JCOMM).


| Extreme Climate Indices | Definition | Unit | Trend - Mann Kendall test (tau) | | |
|---|---|---|---|---|---|
| | | | Winter | Spring | Summer |
| **Intensity** | | | | | |
| Min Tmax | Coldest avg. daily max. temp. | o C | 0.02, p =0.8 | -0.02, p =0.87 | 0.09, p =0.45 |
| Min Tmin | Coldest avg. daily min. temp. | o C | 0.11, p =0.3 | 0.02, p =0.86 | 0.03, p =0.82 |
| Max Tmax | Warmest avg. daily max. temp. | o C | 0.05, p =0.6 | -0.03, p =0.79 | **0.33, p <0.01** |
| Max Tmin | Warmest avg. daily min. temp. | o C | -0.05, p =0.6 | 0.04, p =0.73 | 0.11, p =0.36 |
| Diurnal temperature range | Mean difference between daily max. and daily min. temp. | o C | **0.29, p =0.01** | -0.04, p =0.72 | 0.04, p =0.70 |
| AFDD <0 | Accumulated freeze degrees on days where avg. daily air temp. was less than 0°C | o C | **-0.42, p <0.01** | 0.04, p =0.71 | |
| Freeze thaw days (Tmax >0 Tmin <0) | The No. of days when the avg. daily max. air temp. was more than 0°C and the avg. daily min. air temp. was less than 0°C | Days | | -0.04, p =0.70 | |
| **Duration** | | | | | |
| Growing season length | No. of days between the first occurrence of six consecutive days with Tmean >5°C and first occurrence of consecutive 6 days with Tmean <5°C | Days | | | 0.2, p =0.01 |
| Cold spell duration indicator | No. of days with at least six consecutive days when Tmin < 10th percentile | Days | **-0.26, p =0.03** | 0.05, p =0.70 | -0,006, p =0.97 |
| Warm spell duration indicator | No. of days with at least six consecutive days when Tmax > 90th percentile | Days | -0.09, p =0.4 | -0.06, p =0.63 | -0.03, p =0.82 |
| **Frequency** | | | | | |
| Cool days | Share of days when Tmax < 10th percentile | %Days | -0.01, p =0.94 | -0.06, p =0.60 | 0.17, p =0.17 |
| Cool nights | Share of days when Tmin < 10th percentile | %Days | **-0.24, p =0.04** | -0.07, p =0.64 | **0.28, p =0.02** |
| Warm days | Share of days when Tmax > 90th percentile | %Days | -0.01, p =0.94 | -0.08, p =0.60 | 0.21, p =0.09 |
| Warm nights | Share of days when Tmin > 90th percentile | %Days | 0.09, p =0.48 | -0.08, p =0.50 | 0.22, p =0.07 |
| Frost days | No. of days when Tmin < 0 | Days | **-0.29, p =0.01** | -0.10, p =0.43 | -0.08, p =0.56 |
| Icing days | No. of days when Tmax < 0 | Days | -0.23, p =0.07 | -0.10, p =0.42 | |
| **Precipitation** | | | | | |
| **Intensity** | | | | | |
| Max 1-day precip. | max. 1-day precip. total | mm | -0.08, p =0.54 | -0.11, p =0.42 | 0.09, p =0.44 |
| Max 5-day precip. | max. 5-day precip. total | mm | 0.05, p =0.67 | -0.13, p =0.32 | |
| Simple daily intensity index | Total precip. divided by the No. of wet days (i.e., when precip. > 1.0 mm) | mm | 0.02, p =0.83 | 0.15, p =0.27 | 0.17, p =0.16 |
| Contribution from very wet days | Sum of daily precip. > 99th percentile | mm | 0.05, p =0.66 | 0.15, p =0.27 | 0.05, p =0.68 |
| Contribution from wet days | Total precip. from days >1 mm | mm | -0.13, p =0.33 | -0.17, p =0.16 | 0.12, p =0.34 |
| **Duration** | | | | | |
| Consecutive wet days | max. No. of consecutive wet days (i.e., when precip. >1 mm) | Days | 0.04, p =0.75 | -0.21, p =0.17 | |
| Consecutive dry days | max. No. of consecutive dry days (i.e., when precip. | Days | -0.20, p =0.11 | -0.21, p =0.17 | 0.14, p =0.28 |
| **Frequency** | | | | | |
| Heavy precip. days | No. of days when precip. >10 mm | Days | | | 0.10, p =0.46 |
| Very heavy precip. days | No. of days when precipitator >20 mm | Days | 0.10, p =0.42 | | |
| Total snow days | No. of day when TA> 0 | Days | **-0.38, p =0.002** | | |
| **Runoff** | | | | | |
| Winter Qmin (1981–2022) | Min. winter runoff | mm day⁻¹ | **0.31, p =0.004** | | |
| Summer Qmin (1981–2022) | Min. summer runoff | mm day⁻¹ | | | -0.03 p>0.1 |

27 Extreme Climate change indices

Extreme Climate change indices

**Extreme climate indices**

Trends in extreme indices during the winter suggest that winters are becoming warmer with significant changes in diurnal
temperature range, AFDD <0, cold spell duration, cool nights, frost days and total snow days (Table 1). The extreme climate
change indices during spring showed no significant trends (p >0.05) across the 30-year period. Analysis of summer seasons
showed significant increases (p < 0.05) in the MaxTmax (warmest average daily maximum temperatures) and growing season
length (Table 1, Fig. S6). No significant increase in precipitation indices was found during summer.

**Variability in seasonal runoff data**

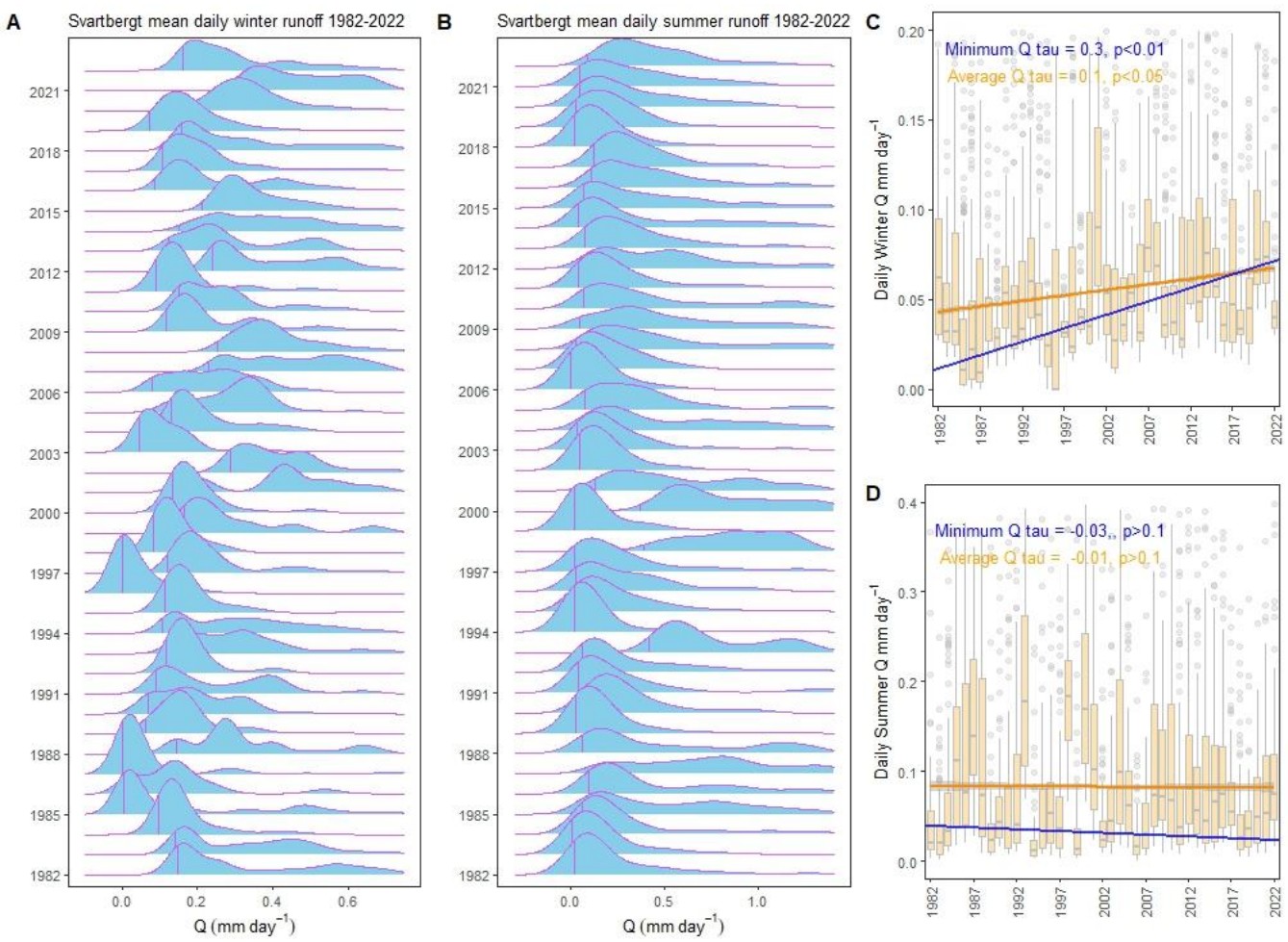

**Figure 5 Variability in seasonal runoff showing hydrograph during the winter (A) and Summer (B) in the C7 catchment. The
minimum flow in each year during the winter and summer are identified as vertical lines each year. The annual variability of
winter runoff is shown as boxplots (C) and trends in minimum (blue line) and average (orange) runoff across the years. In panel D,**

**the boxplots show the annual variability of summer runoff (D) and trends in minimum (blue line) and average (orange) runoff across the years.**

Looking at the runoff time series between 1982-2022, we observed similar increases in winter trends in Qmin (tau = 0.3 p<0.01). Winter Qmin varied between 0.001-0.05 mm day$^{-1}$ where on average runoff ranged between 0.1 mm day$^{-1}$ and 1.1 mm day$^{-1}$. The lowest runoff was recorded in 1996, and the highest runoff was recorded in 2007 (Fig. 5C). During the summer, the minimum trends in runoff were also decreasing (tau = -0.01, p=0.14) with average daily runoff varying from 0.13 mm day$^{-1}$ to 2.45 mm day$^{-1}$. Daily minimums varied from 0.01 mm day$^{-1}$ to 0.4 mm day$^{-1}$ while maximum runoff during the summers varied from 0.8 mm day$^{-1}$ to 21.9 mm day$^{-1}$. The driest years were recorded in 2006 where the average daily minimum was 0.03 mm day$^{-1}$, and the wettest years were recorded in 1993 where the average daily minimum was 0.42 mm (day$^{-1}$ (Fig. 5D).

**Model of changes in runoff**

To understand which hydroclimatic variable was driving the minimum runoff during the winter and summer seasons, we use a stepwise linear regression analysis to identify the best explanatory climate variables of the succeeding season for runoff in Minitab Statistical Software 17. A trend detection test was first performed in R to determine if there was a significant trend in each seasonal data set using the trend package and Mann Kendal trend test. Only the extreme climate variables that showed a significant trend (p <0.05) were used in the seasonal runoff regression models.

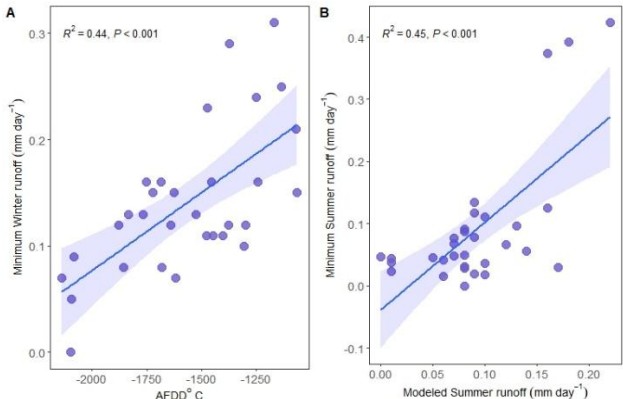

**Figure 6 The best explanatory climate indices of winter minimum runoff (Qmin) were the accumulated freeze degree days below zero (AFDD<0) (W_Qmin=1.26 - 0.0065 AFDD <0) for the winter and the summer (S_Qmin= 0.3 - 0.007 S_Max Tmax + 0.00004 W_AFDD <0).** The standard error in each dataset is represented by the blue shades.

From this analysis, we found that the winter accumulated degree days below zero (AFDD<0) were the best explanatory variable of Qmin during the winter ($r^2$=0.44 p<0.05, W_Qmin=1.26 - 0.0065 AFDD <0) (Table S1). The analysis showed that winters with fewer AFDD<0 (warmer winters e.g. 2014 when AFDD was 1053 °C), were associated with higher winter Qmin (0.2 mm day $^{-1}$) while winters with more AFDD<0 (cold winters e.g. 1994 when AFDD was -2138 °C) showed low winter Qmin (0.1 mm day $^{-1}$) (Fig. 6A). The best explanatory factor of runoff variability in summer Qmin was a multivariate model (S_Qmin=

0.3 - 0.007 MaxTmax + 0.00004 W_AFDD <0) which included winter variables W_AFDD<0 and Summer MaxTmax ($r^2$=0.45

p<0.05) as found from the stepwise linear regression model. The three years (1993, 1998 and 2000) were the largest outliers

in the model because these were the wettest years when minimum runoff was above 0.35 mm day$^{-1}$ (Fig. 6B).

**Isotope analysis using season origin index (SOI)**

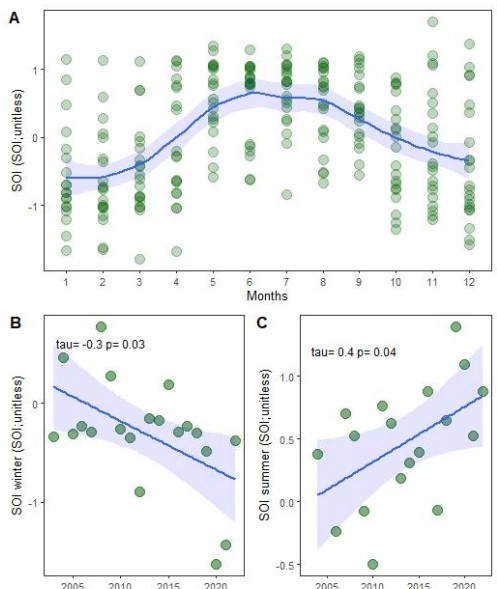

**Figure 7 Variability in season origin index (SOI) daily values from 2003-2022 in the C7 catchment with a loess curve (blue)**
**representing average monthly SOI values (A). The fraction of midwinter precipitation and midsummer precipitation that becomes**
**stream flow each year is shown in panels B and C fitted with linear regression.**

To test if the changes in runoff identified during the winter and summer seasonal analysis of Qmin could be explained by the

contribution of seasonal precipitation to stream flow, we use the SOI analysis to further understand observed trends. This

analysis showed that the SOI in the C7 catchment showed strong seasonality across the years when looking at monthly

variability. In the colder months, SOI averaged -0.5 in December, January and February while in the warmer months, averages

increased to 0.5 in June, July, and August (Fig. 7A). A closer look at the winter SOI in January showed that a greater fraction

of the winter precipitation was represented in the stream flow across time as indicated by the SOI values shifting closer to -1

(typical of mid-winter precipitation) (Fig. 7B). This corresponded with the higher Qmin during the winter identified using the

runoff analysis. The SOI in the summer (July) increased closer to 1 (typical of midsummer precipitation) indicating that a

larger fraction of summer precipitation in later years became stream flow relative to the total precipitation suggesting a shift

away from the mid-winter signal (-1) (Fig. 7C). This infers a lower contribution of winter precipitation to summer stream flow

corresponding to the decreasing trends in Q min during the summer identified using the Qmin trend analysis.

**Discussion**

Analysis of extreme climate indices over the past 40 years in the boreal Krycklan catchment has shown that the most
pronounced changes occurred in the winter where warmer winters have increased minimum runoff quantities during the winter,
which resulted in an exhaustion of water left for baseflow in the succeeding summers. The changes in runoff during the winter
and summer are further supported by isotope analysis that shows an increasing contribution of winter precipitation to winter
runoff and a decreasing contribution to summer runoff. These results based on a 40-year time series, are consistent with a
longer-term trend dating back to 1891 from a nearby meteorological station (Stensele), which shows an acceleration,
particularly in the last decades (Fig 2A, 4A and B). Nevertheless, the 40-year time series of both meteorological and
hydrological data allowed studying the effects of changing seasonal hydro-climatic variables on runoff in ways that previously
had not been possible. The findings presented in this study suggest that warmer winter climates provide direct feedback on the
hydrological flow regime with consequential changes to the seasonal distribution of water in the catchment linked to the
inferred change in water storage. This reconfiguration of flow pathways may reduce subsurface water storage capacity, with
potential consequences for aquatic ecosystems, soil moisture retention, and the resilience of water supplies during dry periods.
In boreal catchments, snowpack typically acts as a natural reservoir, accumulating water through the winter and gradually
releasing it during the spring melt (Trenberth 2011). Under historically colder conditions, this process ensured a slow recharge
of both surface and subsurface waters. However, our results show a shift towards warmer winter conditions, as evidenced by
six temperature and snow-related climate indices (Table 1, Fig. S6). This warming corresponds to a significant reduction in
the number of snow days and increased frequency of mid-winter thaw periods, consistent with regional and global studies
(Sillmann et al. 2013, Easterling et al. 2016). Isotopic analysis of $\delta^{18}O$ further confirmed that the increased winter Qmin is not
primarily driven by higher total winter precipitation but rather by enhanced mid-winter snowmelt, which rapidly delivers
precipitation to streamflow instead of storing it as snow. Supporting this, three supplementary analyses found no significant
relationship between autumn conditions and mid-winter runoff (Fig. S4), suggesting that the dominant control is winter
temperature, not antecedent moisture conditions. The results indicate a climatic shift in hydrological partitioning, favouring
rapid surface runoff over groundwater recharge and sustained summer streamflow.

Modelling the winter Qmin runoff using a stepwise linear regression showed that the best predictor was the AFDD<0 (Fig.
6A), which illustrated those warm winters with lower AFDD<0 had higher winter Qmin compared to the years with colder
winters. These results can be explained by an increase in the frequency of mild (>0 °C) periods and/or induced mid-winter
snowmelt as suggested by (Laternser and Schneebeli 2003), as compared to colder winters that produce deeper snow cover
and delayed snowmelt (Bokhorst et al. 2016, Rixen et al. 2022). A similar increase in winter runoff has also been recorded in
Finland (Kasvi et al. 2019, Rutgersson et al. 2022) and in southern Sweden, with earlier lake ice break-up (58 %) between
1913–2014 (Arheimer and Lindström 2015, Hallerbäck et al. 2022). The consequences of warmer winters have already been
found to reduce soil frost (Easterling 2002, Friesen et al. 2021, Girardin et al. 2022), alter stream runoff and timing of
succeeding spring floods (Blöschl et al. 2017, Breton et al. 2022). If such conditions become more frequent, future changes in

the magnitude and intensity of flooding caused by rapid snowmelt due to warmer winters could increase economic uncertainties for vulnerable areas and infrastructures (Tabari 2020, Nasr et al. 2021) should these events become more prevalent.

The best model of summer Qmin runoff was a multiple regression of two indices (Winter AFDD<0 and Summer MaxTmax) ($r^2$=0.45 p<0.05) (Fig. 6B) indicating that both warmer winters and warm/dry summer conditions affect low runoff in the
summer. This is in line with previous research that showed the importance of antecedent hydro-climatic conditions in explaining the inter-annual variability in summer runoff (Earman et al. 2006, Beaulieu et al. 2012, Van Loon and Laaha 2015, Dubois et al. 2022, Kinnard et al. 2022). Analysis of the SOI-based on $\delta^{18}O$ data showed a distinction between summer and winter values (Fig. 7A). Based on summer SOI values from 2002 to 2022, we observed increasing positive values across time indicating a reduction in winter precipitation in summer stream flow (Fig. 7C). The trajectory of decreasing winter precipitation
infers that there is reduced contribution from winter/spring groundwater recharge in summer stream runoff. The physical mechanism that facilitates this could be related to mid-winter snowmelt where a greater proportion of snow melts during winter, which shifts the seasonality of winter streamflow to an enhanced winter precipitation signal (Fig. 7B) resulting in less water available for aquifer recharge as seen across other cold regions (Jenicek et al. 2016, Boumaiza et al. 2020, Teutschbein et al. 2022). With the higher rates of evapotranspiration observed across the years (Fig. 3A), warm winters together with high
summer temperatures, intensify the magnitude of low flows in boreal catchments. While these findings highlight the vulnerability of groundwater resources in Fennoscandia to changes in winter and summer climate conditions, they call for a deeper understanding of how the changes in the magnitude, intensity and frequency of other hydrological events will be affected in the future.

To understand whether the observed climate change in this study was driven by much larger climate systems, we investigated
the effects of the North Atlantic Oscillation (NAO) index on the winter and summer conditions (Fig. S7). While previous studies have shown a connection between the NOA (Ulén et al. 2019) and AMOC (Schenk et al. 2018) to winter climate conditions (winter temperature, precipitation, snow accumulation, frozen soil) in southern Sweden and along the coast of Norway, no such effects have to the best of our knowledge previously been reported related to the climate in the north. However, using the NAO index (https://www.ncei.noaa.gov/access/monitoring/nao/) to test the correlation with seasonal
hydro-climatic variables from the Krycklan catchment, we found significant correlations with four winter indices (average winter temperature, Tmin, cold spells and warm spells) and four summer temperature extreme indices (average summer temperature, Tmax, summer cold spells, summer cold days) (Fig. S7). These results suggest the observed changes in winter and summer climate could related to the much larger regional NAO system, which in turn can be key to understanding the mechanism regulating winter runoff processes.

The findings of this study emphasize the substantial implications of warm winters for hydrological processes within boreal catchments. Specifically, the observed increase in winter runoff, coupled with the altered timing and magnitude of the cold season, signals significant changes in water storage and redistribution over the course of the year. Notably, the intensified mid-winter snowmelt events appear to reduce the snowpack that would otherwise contribute to spring runoff and groundwater recharge, diminishing water availability during the critical summer months when baseflow and groundwater are most critical

(Jasechko et al. 2017, Klove et al. 2017, Nygren et al. 2020). A higher risk of hydrological droughts could occur where deficits in both surface and subsurface water coincide resulting in stronger impacts on biophysical conditions and biogeochemical processes (Blahušiaková et al. 2020, Teutschbein et al. 2022, Bouchard et al. 2024), and hence aquatic organisms (Williams et al. 2015, Kreyling et al. 2019). While the isotopic analysis confirmed shifts in the relative contributions of winter precipitation to streamflow, it also highlighted the complexity of these processes. The increased winter runoff does not directly

correlate with increased winter precipitation, suggesting that warmer winter temperatures and mid-winter snowmelt are the primary drivers of these changes. These results point to the potential for more frequent and severe hydrological extremes, such as flooding from rapid snowmelt combined with rain on snow in winter and water scarcity in summer, as winter warming intensifies. During the low flow regime, the strong control of stream temperature affects life in aquatic ecosystems as well as the water availability for drinking and irrigation purposes, exacerbating low reservoir levels and decreasing hydropower

generation (Dierauer et al. 2018). Other implications of warmer winters and summer processes have already been shown to affect net ecosystem exchange (NEE) (Monson et al. 2005), water table depth (Nygren et al. 2021, Dubois et al. 2022, Dao et al. 2024), ecological processes (Hrycik et al. 2021), and tree growth decline (Laudon et al. 2024). While higher winter runoff moderates snowmelt-related flooding (Blöschl et al. 2017, Irannezhad et al. 2022), lower groundwater recharge can result in drier summer landscapes with severe consequences on wildfire activity (Westerling et al. 2006), tree mortality (Sterck et al.

2024), ecosystem stress (Hatchett and McEvoy 2018) and carbon uptake (van der Woude et al. 2023). These implications underscore the urgent need for further research on how changes in seasonal runoff dynamics, driven by broader climatic factors like the NAO, will impact regional water resources and ecosystem functions in the future.

**Conclusion**

In this study, we found significant trends in many warming-related climate extreme indices over the last four decades during

both winter and summer. The warming observed in the last 40 years corroborates with the longer 130-year time series that dates to the 1890s showing progressive warming temperatures across time. Evaluating how these changes affect key hydrological processes during the same period, we observed higher runoff during the winter while decreasing runoff during the following summers. Using the significant trends in the extreme indices to evaluate the effects on catchment seasonal runoff, we found that winter variables were best at explaining winter minimum runoff while winter and summer maximum

temperatures could explain the changes observed in summer minimum runoff. These findings were supported by water isotopic analysis that showed an increasing seasonal origin index during the winter indicating higher contributions of winter precipitation to winter runoff and consequently lower winter precipitation to summer runoff. With the decreased catchment water storage due to increased winter runoff before the occurrence of the true spring flood, the potential for maintaining summer baseflow runs the risk of being exhausted in the future, should warming trends persist. This work highlights the

importance of understanding future hydro-climatic trends where changes in the seasonal distribution of water could further affect low-flow conditions, with implications for drought-related issues considering future climate change.

**Code Availability and Data Availability**

Runoff data, meteorological data and isotope data for Svartberget can be downloaded from the SITES data portal (https://data.fieldsites.se/portal/). Long-term data from the Stensele station can be obtained from SMHI achieves. Codes used in the production of graphs and their associated files can be found on fig.share.com https://doi.org/10.6084/m9.figshare.27320538

**Author contribution**

HL developed the initial concept assisted in the analysis of results and created the first draft of the manuscript. TT processed the data, analysed the results and wrote the first draft of the manuscript.

**Competing interest**

The contact author has declared that none of the authors has any competing interests.

**Special issue statement**

This article is part of the special issue "Northern hydrology in transition – impacts of a changing cryosphere on water resources, ecosystems, and humans (TC/HESS inter-journal SI)".

**Acknowledgement**

We would like to thank the staff from the SLU Unit for Field-based Forest Research for technical and logistic support.

**Financial support**

The study was funded by the Knut and Alice Wallenberg Foundation (grants 2018.0259; 2023.0245). The study site Svartberget is part of the Swedish Infrastructure for Ecosystem Science (SITES) and the Swedish Integrated Carbon Observation System (ICOS-Sweden) research infrastructure. Financial support from the Swedish Research Council and contributing research institutes to both SITES and ICOS-Sweden are acknowledged.

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
