# Peer review of "Trends in hydroclimate extremes: How changes in winter affect water storage and baseflow"

_Hydrology and Earth System Sciences, 2024_

## Author Comment (AC2)

**Response to Reviewer 2:**

Comments/Text of reviewer posted in **black;** our answers are posted in blue.

**RC2**:

This manuscript examines long-term trends and variability in the seasonal and annual temperature, precipitation and runoff in the Svartberget catchment, within the Krycklan catchment, located in the northern Sweden's boreal zone. This manuscript compliments many of the past studies done within this catchment, adding a much-needed analysis on the long-term effect of changes in temperature and precipitation on the runoff.

Overall, the manuscript is well written and well structured. I believe that the current manuscript requires additional analyses and information to be publishable as a journal article in Hydrology and Earth System Sciences. Provided below are a list of questions, comments, and suggestions towards an improved version of the manuscript.

Response: We thank the reviewer for the thorough review and positive feedback on this research. We appreciate the detailed comments and suggestions for improvement of the message and analysis. We will address each point as detailed below.

General Comments:

> 1a) Line 85: With the hypothesis that warmer winters will result in higher runoff during the winter, exhausting summer baseflow, how much more runoff is expected to occur during your defined winter period from snowmelt? If temperatures remain below freezing during the winter, then I suspect an increase in winter baseflow would be from the previous autumn season.

Response: In response to "how much more runoff is expected to occur during your defined winter period from snowmelt", we can see that using the accumulated freeze degree days (number of days less than 0; AFDD) between the period 1992-2022 on average was $-1579^{o}C$ with an average winter runoff of 0.13 mm day $^{-1}$. In the colder years (such as 1994 and 2003) with an AFDD of below -2000 $^{o}C$ we observe the lowest Q (below 0.1 mm day $^{-1}$). Contrary during warmer years with AFDD above -1250 $^{o}C$ (such as 2015) we observed significantly higher runoff (commonly above 0.2 mm day $^{-1}$) where significantly higher runoff was observed over time ($r^2$= 0.25, p<0.01 Fig 3B). We will add this to the results to improve the novelty of this research.

[Figure]

Fig 1 Change in winter runoff in relation to Accumulated freeze degree days over time

1b) Would the changes in the timing of snowmelt most likely affect the spring season thus creating more low flow events in the summer?

Response: Analysis of the initiation day of melt during the spring (day that runoff increased above baseflow) showed that snowmelt occurs earlier over time (as shown using the Mann Kendal trend test (tau =-0.29 p=0.02  Fig 4A)). Using regression analysis to detect the relation between winter extreme indices and spring initiation day of melt, we note that winters with a higher number of frost days correlated with earlier spring melt, which supports that less water is available in the catchment for summer low flow (Fig 4B). However, since we focused only on minimum runoff indices and extreme climate indices, including other runoff indices is beyond the scope of the manuscript.

[Figure]

Fig 2 Analysis of changes in the timing of spring melt (A) and its relation to winter climate (B)

2) Line 126: Why was the autumn season not used in this study? Late autumn discharge, enhanced by autumn rainfall and early snowmelt sessions, will influence the total winter flow. An analysis of the autumn season should not be omitted in this study.

Response: Following the suggestion of reviewer 1 (and this comment), we have tested the effect of autumn precipitation and runoff conditions on winter runoff (see reviewer 1 comment 2) using three techniques. The results did not show any significant relation between Autumn variables and winter runoff. However, we do appreciate this question in testing the validity of the winter runoff driven by winter snowmelt rather than stored water from previous autumn events. This information will be added to the manuscript.

3) Are there any strong connections of climate oscillations (e.g. AO or NAO) to the climate variables or climate indices that could then influence runoff in this catchment?

Response: While previous studies have shown the connection between North Atlantic Oscillation (NOA; Ulen et al 2019) and AMOC (Schenk et al 2018) to winter climate (winter temperature, precipitation, snow accumulation, frozen soil) in South-western Sweden and along the coast of Norway, no such effects have to the best of our knowledge been observed related to the climate in Northern Sweden. However, using the NAO index (https://www.ncei.noaa.gov/access/monitoring/nao/ ) to test the correlation with seasonal hydro-climatic variables from the Krycklan catchment, we found significant correlations with four winter and four summer temperature extreme indices. Although we did not find any direct connection to runoff or precipitation variables during the seasons, these new results do suggest that NAO can have an important role in regulating local temperatures in the Krycklan catchment, which in turn can drive processes regulating winter runoff. These new results will be incorporated into the discussion section to support the effect of climate change on NAO and local climate in Northern Sweden and the figure will be added to the supplementary figures.

[Figure]

Fig 3 Regression of winter and summer variables with the North Atlantic Oscillation Index (NAO) during the winter and summer

4) In order to detect trends and significant changes, I am wondering why a more suitable Mann-Kendall test and Sen's slope was not employed on the time series?

Response:  Indeed the non-parametric Mann-Kendall test was first performed for detecting trends in the dataset during the initial analysis (see also comment by Reviewer 1). However, we did not include the results in the submitted version of the manuscript but have done so now following this comment. It should be noted that the results did not change by including the Mann Kendal test other than by improving the prediction of trends in some cases. All figures will be updated with the tau values and a table has been added to supplementary information containing all trend values.

5) Any map that you could provide for the readers of the Krycklan and Svartberget catchment, including location of sampling sites, met tower(s), and hydrometric stations?

Response: The map will be added following this and the other reviewer's suggestion

[Figure]

R Fig 4 Map showing the location of the study site in the Krycklan catchment in relation to Northern Countries

6) Have you considered using a change-point detection on the time series to detect any regime shifts?

Response: Following this comment, we checked for change points using r statistical software change point package for the Runoff, Average Winter Temperature, Precipitation and Evapotranspiration using the local climate data (1982-2022). No change point was detected in the average temperatures, however, we did identify change points in Q, P and ET. These change points did not coincide, and instead occurred at different times for each variable (Q in 1998, P in 1994 and ET in 2005) and therefore could not provide any additional information that could help in understanding the connection between winter climate and seasonal runoff.

[Figure]

Fig 5 Change point detection analysis using R studio strucchange package for the Svartberget annual data on Evapotranspiration (ET), Annual Runoff and Annual precipitation

**Specific Comments:**

1) Page 1, Title: In the manuscript, you focus on the other seasons, not just winter. I suggest changing the title to reflect your analysis.

Response: We appreciate the comment but would like to emphasize that we are interested in showing the effects of winter conditions on successive seasons and therefore would keep our original title.

2) Line 105: What is the difference between the two different data sets? Was the shorter period data set only missing daily maximum and minimum temperatures? Why not use the 1982-2022 data set to detect long-term trends and "extreme climate change"?

Response: Indeed the difference in the dataset was the maximum and minimum temperatures. Since the extreme indices used Max and Min to define each index, we were not able to calculate indices using the longer dataset. We will make this clearer in the next version of the manuscript in the method description section.

3) Line 110: What is the difference between the two stations? Was any analysis conducted to see the similarities between the two locations that are 150 km apart?

Response: The data from the Stensele and Svartberget synchronise with an $r^2$=0.92, root mean square error (RMSE 0.05) for the overlapping period 1982-2004 using the daily average time series. The systematic bias in the dataset lies in the extremes where maximums are higher in the Svartberget than the Stensele dataset while minimums are lower in the Stensele than the Svartberget dataset. However, these differences are small and do not affect the general trends in the dataset used in this study. Based on these results, we expect that the trends observed in the long-term Stensele dataset can be used to reflect similar trends in Svartberget. Thank you for the comment, we have included this in the supplementary information section to support the data analysis.

[Figure]

Fig 6 The correlation between daily mean temperature from the Stensele meteorological station and the Svartberget meteorological station between 1982-2004 showing the $r^2$ and RMSE (root mean square error) between the two datasets

4) Line 116: What percentage of the data set was gap-filled and the quality of the data? The reference, Karimi et al. (2022), does not mention this information for the 1982-2022 time period.

Response: Gap filling was done for 4% of the dataset as described in Karlsen et al 2016. Thank you for the comment; we will add this to the method description to improve the clarity of the dataset limitations.

5) Line 144: Do you avoid possible inhomogeneity as described in Zhang et al. (2005)? Zhang, X., Hegerl, G., Zwiers, F.W. and Kenyon, J. (2005) Avoiding Inhomogeneity in Percentile-Based Indices of Temperature Extremes. Journal of Climate, v18, 1641-1651. DOI:10.1175/JCLI3366.1.

Response: To address inhomogeneity in the dataset, we used the bootstrap technique to test if the trends in climate indices would vary depending on the window use ie 3 days, 5 days, 7 days) based on the description of the climate index (https://etccdi.pacificclimate.org/list_27_indices.shtml). Since these data were aggregated by seasons and not by weeks or months, the windows did not cause the results to vary so we used the definition as published to be consistent.

6) Line 144: For Table S1, could you provide more details on how these are calculated?

Response: We will add a link to the 27 extreme climate indices that describe in detail how each index was determined and will add a short description to Table 1 in the manuscript to improve the clarity of how the extremes were determined.

7) Line 146: For the maximum 1-day precipitation total, what if the precipitation event starts at night and ends in the morning? How do you account for this?

Response: We used a 24-hour technique where the day starts at midnight 00:00 and ends at 23:59. Within this period, all recorded precipitation was summed to reflect the daily total. If a precipitation event started before 00:00 in the previous evening, it was recorded in the previous day's daily sum. We assume that even if we missed an

extreme in precipitation because of how a day was defined in maximum 1-day precipitation, we expect that this would be reflected in other indices such as maximum 5-day precipitation, simple precipitation index etc.

8)  Line 175: Define SOI as the Seasonal Origin Index here.

Response: We will add the definition of SOI to this section based on this suggestion

9)  Line 191: The change in average daily temperatures from two different years is misleading and is best to just stick to the slope value of 2.2. Similar comment for the select years on Page 8, unless you are mentioning the upper and lower boundary of the variability.

Response: We agree and will use the long-term trend values instead of the changes in the years

We now suggest formulating this as " In the last four decades, the increase in temperature has accelerated by 2.1 °C where long-term average daily temperature trend changed from 1°C in the 1980 to 3.1 °C in 2022".

We now suggest formulating this such as "A closer look at the last 30 years of variability in temperatures in the Krycklan catchment (1992-2022) showed annual average temperatures ranging from 0.5 to 5 °C across the years increasing from 1.5 °C in 1992 to 3.1 °C by 2022 (Fig 1B) when looking at the long-term trend line."

10) Figure 1B: What does the grey shaded area represent on these scatter plots?

Response: The grey shaded area represents the standard error in the dataset. This info will be added to the caption.

11) Figure 1C: What does the blue shaded areas represent? Are the numbers 1-12 suppose to represent each month? The caption for Figure 1 in general is very confusing and jumps between Figure A, B and C.

Response: We will restructure the caption to improve clarity as follows

We now suggest formulating this as "Figure 1 Trend in long-term temperature in the Krycklan catchment showing the relation to the much longer-term time series from the SMHI-200 Stensele data (1891-2004) (A), trends in average daily air temperature (Avg), maximum (Max) daily air temperature and minimum (Min) daily temperatures, and variability in daily temperatures in the months across the year (C), across 30 years from 1992-2022. In panel A, the Red symbols indicate the average within a 30-year period from 2022 backwards while purple and orange error bars represent the standard deviation in the Stensele and Svartberget datasets, respectively. The grey shaded area in panel B represents the standard error in the datasets and the blue jitter dots in panel C represent daily average temperatures in Svartberget."

12) Line 207: What is the difference between these minimum temperatures and those in the previous sentence?

Response: In line 206, we show the annual average temperatures, which ranged between 0.5-5 º C and in line 207, we show the annual minimum temperature, which ranged -4.1 to -0.4 º C. We will omit the term "average" in line 207 to improve the clarity of this sentence.

13) Line 208: Was this "coldest month" always the same month?

Response: Looking back at the dataset, we see that across the years, the coldest months were Jan and Feb. We will change this sentence to reflect this.

We now suggest formulating this such as:  "The annual average minimum temperature ranged from -4.1 to -0.4 ºC with the coldest average monthly temperatures occurring in either January or February, ranging from -20.6 to -7.6 ºC (Fig. 1C)."

14) Figure 2: This should be placed after the following paragraph. What are the legend labels in Figure 2A? This is the first time evapotranspiration has been mentioned. I would include a quick comment in the Methods section on how it was calculated. How much confidence is there in these values and the increasing general trend?

Response: Good suggestion, we will add a short description of ET in the methods and have reworded the cation for improved clarity. Based on the Mann Kendal analysis which showed a positive tau and p-value <0.02, we can show high confidence in the ET trends.

We now suggest formulating this as:  "Figure 2 Long-term trends in annual Precipitation (P) and runoff (Q) from the Krycklan catchment across 40 years showing the differences in the percentage of precipitation (P) from runoff (Q) that represents evapotranspiration (ET) (panel A). The variability as measured by the standard deviation in runoff (Q) and precipitation (P) during the time period is shown in panel B. The shaded areas represent the standard error in the datasets. "

15) Line 228: What about the variability in precipitation and its increasing trend?

Response: We will add a sentence to explain the variability in P.

We now suggest formulating this as:  "It should be noted that the variability in runoff has decreased by 0.5 mm day$^{-1}$ from 1.6 mm day$^{-1}$ to 1.1 mm day$^{-1}$ while precipitation increased from 3.5 to 4.2 mm day$^{-1}$ during the period 1982-2022 (Fig. 2B, Table S2, Fig. S2)."

16) Line 233: Is the same method as described in the Methods section? I would delete this sentence or at least make it more clear for Figure S3.

Response: This sentence will be omitted to improve clarity based on the reviewer's suggestion

17) Line 235: What is the difference between "daily averages" and "average winter temperatures" here?

Response: This sentence will be omitted to improve clarity

18) Figure 3: What is the "blue box" described in the caption? What do the shaded parts of the trend lines represent? In panels C and D, it looks like both the trend and variability, with the variability lines hardly noticeable.

Response: The caption will be reworded and the error bars omitted to improve the consistency of the figure.

[Figure]

Fig 7

We now suggest formulating this as:  "Figure 3 Variability in seasonal temperature across 40 years in the Krycklan catchment (orange box) showing changes in temperature during the winter (A) and Summer (B) in relation to the SMHI Stensele longer-term dataset (1891-2004). The variability in annual air temperature (C) and duration (D) is represented by purple symbols for the winter period and green symbols for the summer period. Changes in the start (red) and end (black) of the winter (circle) and summer (diamonds) are depicted in panels E and F, respectively. Standard errors in the datasets (A, B, D, E, F) are shown as grey shading."

19) Line 254: "MinTmin" was shown not to be significant in Table S1.

Response: Thank you for the observation, we have omitted the error.

20) Line 255: Based on Table S1, you have some indices with significant changes, but no significant trends in any of the spring indices or seasonal runoff variables? Is this correct? The same question for Figure 4.

Response: Using the Mann Kendal trend test, we could not detect any significant trends or changes in the spring indices. We have changed the analysis to the Mann-Kendel trend test to make this clearer. In Figure 4 (manuscript), we observe increases in Q during the winter as shown by the positive trend test (tau=0.3 p<0.01). During the summer although we see a decrease across the year, these changes were not significant at p<0.05.

21) Line 262: Are these trends described here different than those listed in Table S1? Seems like you are using two different data sets to examine similar trends. Table S1 is 1992-2022 and this analysis uses 1982-2022.

Response: While we do look at the long-term trends to gain a more robust long-term perspective on how runoff is changing across time as shown in manuscript fig 4, we have been limited to truncating the dataset from 1992-2022 for the modelling analysis based on the availability of extreme climate indices data. Based on this comment, we have added both results to Table S1 and added a sentence to reflect this difference.

We now suggest formulating this as: "Looking at the time series between 1992-2022, we observe similar increases in winter trends tau= 0.3 p<0.01). Winter runoff varied between 0.001-0.05 mm with total season runoff during the winter varying between 18 mm and 144 mm. On average runoff ranged between 0.1 mm day$^{-1}$ and 1.1 mm day$^{-1}$ with the lowest runoff recorded in 1996 and the highest runoff occurring in 2007 (Fig. 4). During the summer, the minimum trends in runoff were also decreasing ($r^2$=-0.05, p=0.14) with average daily runoff varying from 0.13 mm to 2.45 mm between 1982 and 2022. Trends between 1992- 2022 were also consistent with this (tau= -0.08 p=0.14). Daily minimums varied from 0.01 mm day$^{-1}$ to 0.42 mm day$^{-1}$ while maximum runoff during the summers varied from 0.8 mm day$^{-1}$ to 21.9 mm day$^{-1}$. The driest years were recorded in 2006 (0.03 mm day$^{-1}$) and the wettest years were recorded in 1993 (0.52 mm day$^{-1}$) (Fig. 4)."

22) Line 278: Is "MaxTmax" a winter variable or a summer variable?

Response: Thank you for the comment, we will change all indices to reflect S for summer and W for winter to avoid further misinterpretation

23) Figure 6: Any discussion on Figure 6A? Are the numbers on the x-axis representing months?

Response: We will add a citation to where Fig 6A will be explained and the months' label will be added to the x-axis

24) Line 320: The SOI values show an increased contribution from winter precipitation to streamflow. Any reason as to why this does not show up in your analysis of the data sets?

Response: In our analysis, the contribution from winter precipitation to stream flow is reflected in increasing negative values across time. This is indeed shown in Figure 6B where we see SOI shifting from 0.2 to -0.6 during the winter period.

25) Line 321: Were there more mid-winter melting events found in the data sets? Would the shoulder seasons (spring and autumn) be the most affected by the warming rather than winter?

Response: In this analysis, we focused mainly on the lowest flow across the years as an indication of mid-winter melt. Another indicator that can be used is snow depth or soil frost, which also indicates warming. In terms of the effect on the shoulder seasons, we have tested the effects of autumn runoff on mid-winter melt (reviewer 1 comment 2) which did not show any significant correlation with the runoff during the winter. Additionally, for spring, we have found earlier initiation days of melt (not included in this study) in relation to winter frost days (see reviewer 1 comment 1b) indicating that indeed warmer winters are also affecting spring flood dynamics.

26) Line 355: As mentioned previously in the discussion, there was an increased contribution from winter precipitation with the SOI analysis. Figure S4 showed a decrease in total winter precipitation and an increase in total summer precipitation. Any comment on the different results?  How would this influence the probability of droughts in this boreal catchment?

Response: Increased contribution of precipitation to stream runoff during the winter is an indication of increased loss to the catchment through runoff rather than stored in snowpack. This, together with reduced winter precipitation can further induce low groundwater recharge for aquifers (similar to other studies by Nygren et al, 2020, Jasechko et al., 2017, Klove et al 2017). Even though precipitation is increasing during the summer (although not significantly), this may not be enough to recharge the storage or the dried-out landscapes which are water deficit. This is reflected in the low correlation between minimum summer runoff and total precipitation where years with the highest precipitation are not the year with the highest baseflow.

27) Line 372: "Shrinking snowpacks" As mentioned in previous comment, one analysis shows a greater contribution and the other, a non-significant decrease. Did you find that the snowpacks were shrinking?

Response: In this analysis, we have focused on the effects of extreme winter indices, which infers impacts on snowpack indirectly. With the SOI analysis, we showed that the isotopic signature in the stream water during the winter reflects more of the winter precipitation signal across time. With the extreme indices analysis, we have not found any significant increase in contribution from winter precipitation or in the number of precipitation days in the winter. However, the extreme event analysis showed warmer winter temperatures across time, alluding to an increased winter melt of the snowpack. This coincides with the isotopic analysis results.

1) Table S1: What is the difference between Tmin, Tmax and MaxTmax and MinTmin?

Response: Temperature is recorded on a 10-minute interval in the Krycklan catchment where hourly maximum, minimum and averages are calculated. To identify the coldest recorded temperature of the day we used the MinTmin, which selects the lowest hourly recording of each day while Tmin calculates the average minimum temperature of that day. Similarly, MaxTmax identifies the highest hourly temperature recorded in a 24hr period while Tmax reflects the average maximum temperature in a day. We will make this clearer in the description of the methods to improve clarity.

2) Table S1: Which 27 climate change indices were developed by the World Meteorological Organization out of the 33 variables listed?

Response: Great comment, we will separate the 27 indices to improve clarity

3) Table S1: Should the growing season length be under the summer category instead of winter?

Response: We will move the growing season length to summer. Thank you for the suggestion

4) Figure S1: Are the dotted lines the average daily temperature in blue? It is difficult to distinguish between the three data sets in this figure.

Response: We will remake this figure and hope that it is now clearer

We now suggest the following figure:

[Figure]

Fig 8 Daily air temperature showing (A) Hourly averages, (B) Hourly Maximum temperature and (C) Hourly Minimum temperatures

5) Figure S2: For panel C and D, what does the pink shade represent?

Response: We will improve the description of this caption.

We now suggest formulating this such as: "Variability in daily precipitation (A) and runoff (B) and the variability in annual values for precipitation (C) and runoff (D) in the Svartberget catchment. The pink shade represents the standard error in the dataset"

6) Figure S3: If this is average daily temperature for each year, where is the "isolation of the winter period" in each figure?

Response: The winter period would be the area where the average temperature (blue line) falls below the zero line (black). We will add this to the caption and hope that it improves the clarity of this figure.

7) Figure S5: What does the blue shade in the background of each figure represent?

Response: We will add a description to this caption

We now suggest formulating this such as: "The standard error in each dataset is represented by the blue shades."

---

## Author Comment (AC3)

**Response to Reviewer 3:**

Comments/Text of reviewer posted in **black;** our answers are posted in blue.

**RC3**:

**General comments**

The authors studied trends in hydroclimatic extremes using a 40-year time series from the boreal Krycklan catchment in Sweden, in the context of 130 years of climate data from a nearby site. They looked at how different extreme climate indices changed and how they affected seasonal low flows in winter and summer. The authors also identified the best climate indices for predicting both summer and winter minimum flows. The authors have carried out an interesting study that is certainly scientifically relevant. I see the novelty both in the focus on a boreal region, where not so many studies exist and in the robust testing of different predictors of minimum flow. Therefore, I believe that the study has the potential to be published in HESS. However, I have some comments listed below that I would like to be addressed.

Response: We would like to thank reviewer 3 for the constructive comment and the acknowledgement of the novelty of this research. Please see below our detailed response to all the comments raised by the reviewer.

**Major comments**

1) I think the research gaps and novelty could be better described. Although the entire introduction is clear and comprehensive, I miss the "so what" message. What is new in the study and how does it go beyond current knowledge?

Response: In response to the research gaps and novelty, we will add more details to highlight the novelty and gaps in the research as follows in the third paragraph:

We now suggest adding "Despite previous studies showing the effects of earlier snowmelt on spring floods (Irannezhad et al., 2022; Venäläinen et al., 2020; Hrycik et al., 2024), as well as decreasing baseflow trends in succeeding summers (Murray et al 2023), is a link of how winter climate affects winter hydrological processes and subsequent summer runoff lacking. This is critical to understand how climate change will alter winter conditions and consequently catchment recharge and runoff processes in light of future climate warming predictions. "

We will also restructure the final paragraph of the introduction as follows: "This research addresses how changes in winter climate affect seasonal runoff in the boreal Krycklan catchment using trends analysis and interconnectivity with a large number of long-term hydroclimatic variables and isotopic analysis to improve the predictability of future climate warming impacts on hydrological processes. Our first objective was to identify any significant changes in temperature trends and test the relation to a much longer time series (130 years) in the region. We then used a more detailed time series (30 years) to identify seasonal climate extreme indices. With this analysis, we tested how climate is changing using both the 30-year and 130-year time series in terms of seasonal variability. Our second objective was to evaluate how changes in extreme climate indices affect key hydrological processes during winter and summer. This analysis allowed us to quantitatively identify what hydroclimatic extremes are most important for regulating runoff across the seasons and an opportunity to improve the predictability of future climate warming. Our final objective was to use seasonal origin index

analysis of water isotopes to verify whether the changes in runoff corresponded with the climate analysis *by testing the* relative contributions of winter precipitation to winter and summer runoff. "

2) L157: Please describe the linear regression model in more detail. The linear regression models also have some requirements for the input data, such as normality and uncorrelated predictors. Were these conditions met? If not, were the data transformed? Did the authors also consider other methods for detecting trends, such as Mann-Kendall's correlation (which can be used for non-normal distributions thanks to its ranking) or Sen's slopes? How might the interpretation of the results change if different methods of trend analysis were used?

Response: We will change the analysis to Mann-Kendall's trend test, which does not affect the results (see also response to Reviewers 1 and 2). Initially, this test was initially used to identify the significant trends among the indices but was not included in the previous manuscript. However, with this reviewer's suggestion and others, we will include the analysis.

3) L270, section Model of changes in runoff: I think the text in this section should be better supported by the results. For example, the authors state that AFDD<0 is the best explanatory variable, but I cannot see it in any of the figures (Fig. 5 only shows its relationship with winter/summer runoff). Perhaps you could at least show the correlation matrix of all predictors (heat map or similar) to support your statements.

Response: Great suggestion! We will add a correlation matrix to the supplementary material to show the coefficient of regression value with winter minimum Q (WQmin) and Summer minimum Q (S_Qmin). For the winter Qmin, we can see that AFDD had the highest regression coefficient, $r^2$ (0.63) as identified by the stepwise linear regression model. For the summer Qmin, the strongest $r^2$ can be seen with summer max temperature S_MaxTMax (-0.56) and winter AFDD<0 (0.27) indicating that warmer summers lower the Q min while warmer winters contributed to lower Qmin during the summers.

| | $r^2$ | |
|---|---|---|
| Significant climate extreme indices | W_Qmin | S_Qmin |
| W_AFDD <0 | 0.63 | 0.27 |
| W_Frost days | -0.39 | 0.21 |
| W_Icing Days | -0.55 | 0.01 |
| W_Diurnal temp. range | 0.45 | -0.13 |
| W_Coldspells | -0.39 | -0.02 |
| W_Coolnights | -0.45 | 0.03 |
| W_Snowdays | -0.03 | 0.13 |
| S_MaxTMax | | -0.56 |
| S_ Coolnights | | -0.2 |
| S_Warmnights | | -0.05 |
| S_Warmdays | | 0.09 |

Table that will added to the new version of the manuscript.

4) My general concern is that more could be done to investigate the potential effects of individual climate indices on runoff. For example, how does climate variability affect

the observed relationship between climate and hydrological variables, in particular the role of warm/cold, wet/dry or snow-poor/snow-rich years? I believe this direction would provide new insights into catchment behaviour.

Response: We agree that the correlation to years with both temperature and moisture variability with runoff will be an interesting step forward, which could be addressed in a follow-up study, for example using the PLS models to show the most important variables in predicting runoff. However, in this study, the majority of the precipitation variables did not show significant trends, and hence we did not include them in any future modelling steps.

5) While I like the idea of supporting the results of the time series data analysis with a more detailed analysis using stable water isotopes (especially for the Krycklan catchment with long-term and detailed isotope data), I feel that this part is not well connected to the other results and seems quite separate from the rest of the text. In addition, there is only one short paragraph and one figure in the results section relating to this part. I would encourage the authors to expand this part by adding more visualisations and results, and try to better connect it to the results of other analyses.

Response: In regards to the isotope analysis we will improve the text in the introduction as follows

"A powerful technique for assessing changes in stream flow involves using the differences in the water isotopic signatures of $\delta^{18}O$ in precipitation across seasons (Allen et al., 2019). Previously, Peralta-Tapia et al 2016 showed strong seasonality in precipitation signals during the winter and summer where $\delta^{18}O$ isotopes with more depleted (lighter) isotopes in winter min (-24.4 ‰) and enriched (heavier) in summer precipitation max (-4.4 ‰) while stream $\delta^{18}O$ varied between -15.4 ‰ winter and -10.5 ‰ in later summer. This distinction in the seasonal isotopic signals presents the possibility of tracing the fraction of water arriving as snow in the winter that can either be potentially stored in the catchment or become streamflow in the current or proceeding seasons. Similarly, in the summer, the precipitation can be traced to contribute to either storage, streamflow or evapotranspiration (ET). Using isotopic signals to understand the water partitioning in catchments offers a possibility to trace water contributions from different sources that are not possible by using only the water balance or measuring seasonal total precipitation volumes. This can be done by adapting the seasonal origin index (SOI) to implicitly test whether winter precipitation or summer precipitation is overrepresented in stream water with $\delta^{18}O$ isotopes using the methods outlined in Allen et al., (2019). The results from the SOI analysis can then be used to support whether increases in winter runoff are a result of winter precipitation and whether this signal is seen during the summer baseflow. "

We will also add a connection of the runoff analysis to the SOI section as follows.

"The great fraction of winter signal represented in winter runoff corresponds with the results from the runoff analysis that showed higher Qmin during the winter across the years."

**Specific comments**

6) Section 2: It would be good to provide a map of the Krycklan catchment and its position within Europe/Sweden.

Response: A map will be added based on all reviewer's comments.

[Figure]

Fig 1 Map showing the location of the study site in the Krycklan catchment in relation to Northern Countries

7) L110: Is there a reason why the study only uses data up to 2004 for the Stensele climate station? Later, in some figures, the data are plotted together with data from the Krycklan catchment (by different colours). Have the two-time series been homogenised somehow in order to plot them together as a single time series? The distance between Krycklan and Stensele is 150 km, so one would expect different conditions at the two stations.

Response: The meteorological station in Stensele was closed in 2004 and no longer provided data hence we could not continue the long-term analysis with one dataset. We will add this to the methods to make it more clear. Additionally, we have tested the homogeneity of the two datasets using regression analysis and root mean square error (RMSE), which showed that the Stensele and Svartberget synchronise with an $r^2$=0.92, RMSE 0.05 for the overlapping period 1982-2004 using the daily average time series. The systematic bias in the dataset lies in the extremes where maximums are higher in the Svartberget than the Stensele dataset while minimums are lower in the Stensele than the Svartberget dataset. However, these differences are small and do not affect the general trends in the dataset used in this study. Based on these results, we expect that the trends observed in the long-term Stensele dataset can be used to reflect similar trends in Svartberget. We have included this in the supplementary information section to support the data analysis (R Fig 13).

[Figure]

R Fig 2 The correlation between daily mean temperature from the Stensele meteorological station and the Svartberget meteorological station between 1982-2004 showing the $r^2$ and root mean square (rmse) between the two datasets

8)  L134: What is meant by "Temperature intensity"?

Response: Temperature intensity refers to a measure of the degree to which a season is warm or cold as measured by the 7 indices. This is described as exceeding a threshold as described in Sanches et al 2023.  We ´will add this description to improve the clarity.

- the coldest daily maximum temperature (Min Tmax),
- the coldest daily minimum temperature (Min Tmin),
- the warmest daily maximum temperature (Max Tmax),
- the warmest daily minimum temperature (Max Tmin),
- the mean difference between daily maximum and daily minimum temperature (diurnal temperature range),
- the number of days when Tmax < 0°C (icing days), and
- (vii) the accumulated degree days below 0°C (AFDD<0).

9)  L144: Since Table S1 is very important for the methods and results interpretation, I would prefer to put it directly to the main text to avoid jumps between the two files. I think it would be beneficial for the readers.

Response: Thank you for the suggestion, we will move table 1 to the main document

10) L154: How exactly has baseflow been calculated? If baseflow is only represented by minimum streamflow, I would suggest not calling it baseflow (use just minimum streamflow or similar). If you agree, this terminology would need to be changed throughout the text and figures.

Response: Agreed! We will change this to be a minimum runoff

11) Fig. 1: Please explain the grey background colours in the B) (probably confidence or prediction intervals). Same for Fig. 2, 3, 5 and 6. Besides, B) and C) descriptions in Fig. 1 caption are probably switched.

Response: This will be added based on all reviewer's suggestions

12) Fig. 4: Please describe what is in individual panels in the figure caption

Response: We will add labels and describe each plot as follows

"Figure 4 Variability in winter and summer runoff showing hydrograph during the winter (A) and Summer (B) in the C7 catchment. The minimum flow in each year during the winter and summer are identified as vertical lines in each year. The annual variability of winter runoff is shown as boxplots (C) and trends in minimum (blue line) and average (orange) runoff across the years. In panel D, the boxplots show the annual variability of summer runoff (D) and trends in minimum (blue line) and average (orange) runoff across the years.

[Figure]

R Fig 3 New proposed figure

13) Fig. 5: There are a couple of years considerably outside the prediction range. Is there any explanation for what might be the reason for this?

Response: This was explained in line 280:202 in the manuscript "The three years (1993, 1998 and 2000) were the largest outliers in the model because these were the wettest years when minimum runoff was above 0.35 mm day-1 (Fig. 5B)."

14) Fig. 5, caption: I think the resulting equation of the linear model should be placed in the main text together with some further description (see also my major comment related to the linear regression model).

Response: Thank you for the suggestion, we will add the equation to the main text and include further descriptions

15) L298: I would be rather cautious about attributing the results to baseflow, as baseflow was not calculated/simulated in your study (unless I missed something). An increase in Qmin in winter doesn't necessarily mean that baseflow also increases (although I agree that it is likely). If there are more melting periods in winter (higher fast flow component), Qmin will of course also increase, but the effect on baseflow may not be so straightforward. I'm not saying it's not true (as I expect the close connection of streamflow and baseflow in the Krycklan catchment, although I don't know the study area), but I would ask the authors to address this issue in the discussion (e.g. by adding some studies investigating this effect).

Response: Agreed, we will keep this consistent by using minimum flow

16) L324: Again, please consider whether you mean baseflow or Qmin.

Response: Agreed, we will rephrase to be consistent.

17) L354-368: This paragraph is perhaps rather the introduction.

Response: We will consider in cooperation this paragraph in the introduction.

18) In my opinion, the conclusions can be formulated more specifically and with clear take-home messages. Additionally, the sentence in L372 ("shrinking snowpacks that melt earlier") is not a conclusion coming from this study (although the statement itself is true), so please consider reformulation.

Response: We will reword the conclusion based on this suggestion

"In this study, we found significant trends in many warming-related climate extreme indices over the last 30 years during both winter and summer. The warming observed in the last 30 years corroborates with the longer 130-year time series that dates back to the 1890s showing progressive warming temperatures across time. Evaluating how these changes affect key hydrological processes during the same period, we observed higher runoff during the winter while decreasing runoff during the summers (1992-2022). Using the significant trends in the extreme indices to better evaluate the effects on catchment seasonal runoff, we found that winter variables were best at explaining winter minimum runoff while winter and summer maximum temperatures could explain much of the variability in the summer minimum runoff. These findings were supported by water isotopic analysis that showed an increasing seasonal origin index during the winter indicating higher contributions of winter precipitation to winter runoff and consequently lower winter precipitation in summer runoff. With the decreased catchment water storage due to increased winter runoff before the occurrence of the true spring flood, the potential for maintaining summer base flow runs the risk of being exhausted in the future. This work highlights the importance of understanding future

hydro-climatic trends where changes in the seasonal distribution of water could further affect low-flow conditions, with implications for drought-related issues in light of future climate change. "

19) Fig. S2, S5: Please explain the background colours (S2, S5) and the red line (S3) in the figure captions.

Response: These captions will be corrected based on this and the reviewer's 2 comments

**Technical corrections**

20) L8 and 18: Should there be "subsequent" rather than "preceding" (seasons; streamflow)?

Response: Agreed, we will change the terminology to subsequent

21) L111: Please specify the exact time period to be consistent with the previous text.

Response: We will add the runoff timeframe to this section. Thank you for the comment

22) L154-155: Coefficient of determination (rather coefficient of regression).

Response: We will change these to tau using the Mann Kendal trend test based on this reviewer's and others' suggestion

23) L174: The abbreviation SOI should be defined here for the seasonal origin index.

Response: Thank you for the comment, we will add this

---

## Author Response (AR1)

Tejshree Tiwari

Dept. of Forest Ecology and Management
Swedish University of Agricultural Sciences
SE-901 83 Umeå, Sweden.

Dear Editor

Thank you for considering our manuscript titled *"Trends in hydroclimate extremes: How changes in winter conditions affect seasonal baseflow and storage"* by Tejshree Tiwari and Hjalmar Laudon for review in *Hydrology and Earth System Sciences (HESS)*.

We are grateful for the thoughtful comments provided by the three reviewers and the editorial team. In response, we have revised the manuscript thoroughly to address all major concerns and suggestions. Specifically, we have made the following key improvements:

1. Conducted an autumn-season analysis to assess its influence on winter low flow and baseflow conditions.
2. Added a comparison between the Stensele and Svartberget climate datasets to evaluate consistency and robustness.
3. Included a summary table of key results in the main text to enhance readability and accessibility.
4. Applied the Mann–Kendall trend test to all relevant figures to strengthen the statistical evaluation of trends.
5. Improved the clarity and completeness of all figure captions for better interpretability.
6. Incorporated an analysis of the North Atlantic Oscillation (NAO) and its relationship with seasonal climate variability.
7. Expanded and emphasized the role of isotope analysis in understanding seasonal streamflow partitioning.

We believe these revisions have significantly improved the clarity, analytical rigor, and overall contribution of the manuscript. We appreciate your time and consideration, and we look forward to your feedback on the revised version.

Sincerely,
Tejshree Tiwari (on behalf of Hjalmar Laudon)

---

## Author Response (AR2)

**Reviewer's Comment**

**General Comments**

In my opinion, the authors have substantially improved the manuscript, also in response to the comments from other reviewers. They have satisfactorily addressed my earlier suggestions, and I appreciate the inclusion of additional methodological details (e.g., the Mann-Kendall test) and the strengthened interpretation of the results. These revisions have made the manuscript more coherent and convincing overall.

That said, my original major comment (Comment 5) remains. Although the authors have made some modifications to the section on isotope analysis, I still find this part not well integrated with the rest of the results. However, I acknowledge the authors' rationale for not implementing more extensive changes and leave the final decision to the editor.

**Technical Correction**

In one of my comments, I recommended including a correlation matrix to illustrate the relationships among all variables. While the authors agreed in their response, the supplementary material includes only a simple table (Table S2). I understand and accept this decision, but the current caption refers to it as a "correlation matrix," which is not accurate. Please revise the caption accordingly.

**Response to Reviewer**

We thank the handling editor for the opportunity to revise our manuscript and for drawing our attention to the reviewer's helpful comment regarding the caption of Table S2.

In line with Reviewer 3's suggestion, we have updated the caption of Table S2 to more accurately describe it as a table of correlation values, rather than a "correlation matrix." This revision is reflected in the updated supplementary materials.

We would also like to express our sincere appreciation to Reviewer 2 for their thoughtful and constructive feedback throughout the review process, and for their recognition of the improvements made to the manuscript. Their insights have contributed meaningfully to the clarity and coherence of the revised work.